# Recurrent Attention-based Token Selection for Efficient Streaming Video-LLMs

**Vaggelis Dorovatas**[1,2]      **Soroush Seifi**[1]      **Gunshi Gupta**[3]      **Rahaf Aljundi**[1]

[1]Toyota Motor Europe
[2]Archimedes RU, Athena RC
[3]University of Oxford

## Abstract

Video Large Language Models (Video-LLMs) excel at understanding videos in-context, provided they have full access to the video when answering queries. However, these models face challenges in streaming scenarios where hour-long videos must be processed online, and questions need timely responses. In this work, we propose a training-free approach compatible with standard Video-LLMs, leveraging three key concepts: 1) LLM-informed selection of visual tokens to identify those that the LLM has attended to and contributed to its understanding of each short clip. Our attention-based selection allows us to discard up to $\sim 95\%$ of unimportant visual tokens with minimal performance loss; 2) Recurrent processing of past selected tokens to generate temporally coherent understanding of each processed clip; 3) Caption-based question answering for lightweight and accurate responses. Our method achieves state-of-the-art performance on streaming video benchmarks, striking a balance between efficiency and effectiveness.

## 1   Introduction

Efficient and effective video question answering and understanding are crucial for deploying Large Language Models (LLMs) as intelligent assistants in domains requiring continuous visual input comprehension. This capability is essential for applications such as autonomous driving, surveillance, healthcare, and entertainment, where dynamic visual information must be understood in real-time. Current video understanding research with VLMs [31, 17, 2] typically processes entire videos in-context, presenting densely sampled frames alongside text queries to generate responses, with common training approaches focusing on text generation tasks like video captioning or question answering. While effective for short clips, this brute-force approach faces critical limitations with longer videos: computational costs escalate as visual tokens multiply, and exceeding context length limits forces sparse sampling that risks information loss. These challenges become particularly pronounced in streaming scenarios where continuous visual input renders full-video processing unsustainable. This necessitates efficient compression techniques that selectively process frames, condense visual information, and maintain easily retrievable memory structures for effective query response.

Recent efforts have focused on compressing the information from short video clips, moving away from the brute-force approach of processing the entire video in a single pass, and thus extending model capabilities to handle long video understanding. These approaches include methods that either compress only the visual information (before the LLM) encoded by a vision encoder [38, 12, 10], or store only textual descriptions of short clips [1], and retrieve only those relevant to the input

---

*First two authors provide contracted services for Toyota.
*Correspondence: vdorovatas@hotmail.gr

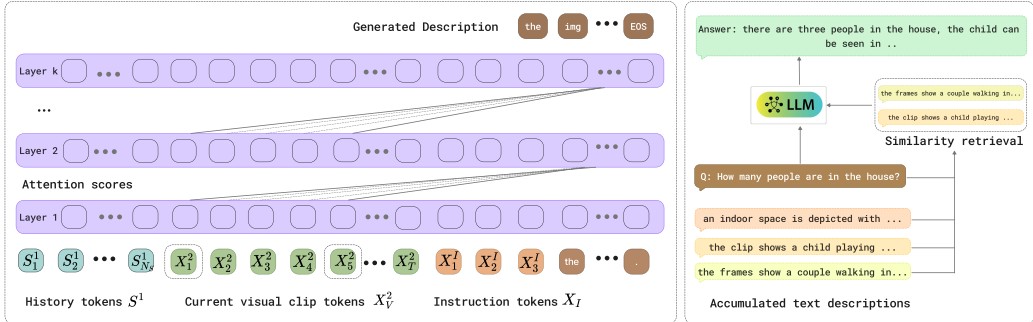

Figure 1: **Pipeline of rLiVS:** (Left): Long videos are broken down into short visual clips which are processed in a streaming fashion. A short clip is pre-pended with selected visual tokens from past clips in the stream to generate a textual description. History is accumulated by selecting a subset of tokens from the short clip based on attention scores. (Right): Given a query on the past video, the accumulated textual descriptions over a long video are compared to the query embedding and the most similar subset is input to the LLM (within the videoLLM) for question-answering.

query. Another line of work involves the LLM in video processing by generating compressed representations in the form of summarization tokens, like VideoStreaming [24], or storing KV-cache during inference, like ReKV [9], effectively combining the two modalities. While all these approaches mark important progress, they face amplified challenges in streaming settings, where past frames cannot be revisited and inputs can span hours. Training-based approaches [24, 38] face extrapolation issues with arbitrarily lengthy videos, and scaling their training can be computationally prohibitive or suffer from data scarcity and annotation issues for long videos. Training-free approaches like ReKV, while avoiding retraining, rely on storing large KV-caches, which becomes memory-intensive and introduces potential redundancy in streaming, ultimately affecting overall performance. Conversely, clip-wise caption-only methods (e.g., Goldfish [1]) offer efficiency but lack continuity, making it difficult to track entities and reason across clips, thus hindering holistic understanding.

In this work, we propose an efficient, training-free, and video LLM-agnostic solution for streaming video question answering. Our approach processes long videos (offline or online) by splitting them into short clips and focuses on three key aspects, integrating visual perception with language understanding for holistic video comprehension: 1) Compressing video information to avoid redundancy and inefficiencies in memory usage by selecting a small set of key visual tokens per short clip, 2) Accessing past selected visual tokens when processing new short clips, enabling visual information sharing across short clips and enhancing overall video comprehension, and 3) Performing text-based retrieval and answering, harnessing LLMs' proven strength in reasoning over extended contexts (such as multi-clip captions).

Specifically, *during the processing* of each short clip (i.e., generating a caption), we draw inspiration from cognitive neuroscience, particularly the interplay between attention and memory [6]. Given memory's limited capacity [23, 7], attention is key to selective encoding [6]. Accordingly, instead of randomly sampling visual tokens, we retain those that received the highest attention from the LLM, treating them as a compressed memory trace of the clip. Furthermore, building on the idea that past experiences shape current attention [6], we reuse previously selected tokens when processing subsequent clips. These past tokens are passed forward with new inputs, allowing memory to guide attention and support context-aware understanding. Inspired by the brain's recurrent visual processing [15], we implement a simple FIFO memory: newly selected tokens are prepended, and the oldest are discarded once the context limit is reached.

Finally, *when answering* a question about past events in a video stream, we retrieve the top $K$ captions based on the similarity of the question to the previously stored captions of short clips. This approach is motivated by the efficiency of storing captions alone and empirical evidence suggesting that long video reasoning is more effective with representative text. This approach enables us to answer questions based on the stored captions, rather than reprocessing the previous visual input, which is a common practice in current works [38, 10]. Figure 1 illustrates our method.

Our approach can be applied to any video-LLM pretrained on short clips, enabling long video understanding without additional training and maintaining a low memory footprint. We evaluate our method on three offline and streaming benchmarks, achieving state-of-the-art performance with significantly lower memory requirements. The simplicity and strong performance of our design set a robust baseline for online long-video question answering.

Our contributions are as follows:

- We propose Recurrent LLM-informed Visual Selection (rLiVS), a simple, training-free approach for long video understanding and question answering.
- Our approach is agnostic to the Video-LLM architecture and does not require any external modules.
- We achieve state-of-the-art performance with significantly lower memory requirements.

In the following, we discuss the closely related work in Section 2 and present our method in Section 3. We evaluate our method and ablate the design choices in Section 4 and conclude in Section 5.

## 2 Related Work

**Video-Language Models:** Recent video-language models have achieved strong performance on short-form video tasks by aligning visual features with text using pretrained vision-language transformers [30, 16, 19]. These models typically sample a small number of frames from each video and encode them into visual tokens using a vision backbone (e.g., ViT), which are then fed into a language model either directly or via cross-modal fusion. However, the number of visual tokens grows linearly with the number of frames and spatial patches—e.g., sampling 32 frames with a ViT-base backbone (16×16 patch size) yields over 1,000 tokens—making it prohibitively expensive to process long videos end-to-end [34, 29]. As a result, these models are limited to short clips (typically under 30 seconds), lack persistent memory, and cannot capture dependencies across longer temporal horizons without substantial computational and memory overhead.

**Offline long-video understanding:** To overcome the limitations of short-clip models, several offline approaches have been proposed for long video understanding by introducing mechanisms to compress or summarize information across time. Some methods focus on compressing only the visual modality, either by keyframe selection, feature pooling, or storing tokenized visual memory [32, 34, 28, 10]. Others rely solely on textual abstraction, converting short clips into natural language summaries that are later retrieved and composed to answer queries [1]. More recently, hybrid methods like VideoStreaming [24] train auxiliary models to generate trainable summarization tokens that condense the content of each clip into a compact representation, which is stored and retrieved during inference. While effective, these methods often require significant offline training to align the summarization with downstream tasks, are tied to specific pretraining pipelines, and typically process each clip independently—limiting their ability to build hierarchical or persistent memory across long videos.

**Streaming and Memory-Augmented Video Understanding**: In contrast to offline approaches, streaming video understanding aims to process incoming video frames incrementally, enabling real-time or low-latency inference over long temporal horizons. Retrieval-augmented methods maintain external memory stores of past embeddings and retrieve relevant context on-the-fly based on the current input, avoiding full-sequence reprocessing [38]. Similarly, MeMViT [35] incorporate visual memory modules or cross-modal attention over cached tokens to improve temporal coherence, though they often rely on fixed-length context windows or soft attention mechanisms, which scale poorly or suffer from memory drift over time. In the same direction of visual memory, Video-XL [27] introduces a custom-trained Video-LLM architecture that learns to compress and retain visual information over time through supervised training. However, its performance is tightly coupled to the specific model architecture and the distribution of video lengths and compression strategies seen during fine-tuning. Recently, ReKV [9] explored streaming video question answering by storing full decoder key-value caches and retrieving relevant context either through internal attention-based mechanisms or external CLIP-based similarity, demonstrating that retrieval over cached embeddings can be both effective and efficient. However, many of these methods either treat memory as a flat buffer or require storing large volumes of raw activations, limiting scalability and longer-term abstraction. Our work builds on these insights by introducing a memory consolidation mechanism that persistently retains and hierarchically abstracts multimodal context across a stream, enabling scalable and task-adaptive long-term understanding.

# 3 Method

We aim to develop an efficient method for streaming scenarios. Our approach uses a short-clip video-LLM as the backbone, which is capable of multimodal processing of short video clips and generating textual responses. Specifically, for the general captioning task, the model, given a sequence of visual tokens (representing the input frames, $V$) extracted from a visual encoder (eg. CLIP [25]) and a potential instruction ("Describe what is happening in the video."), generates a textual description $C$, representing the model's understanding of the short clip. Rather than storing all the information (as in [9]) we aim to compress the visual information by leveraging the relation with the textual tokens of each short clip, producing a compressed visual-guided-by-text representation, $S$. We further target a model-agnostic design differently from [11] and [24]). The key aspect here is that we have the input to the LLM, $V$, and its corresponding caption, $C$, which can be interpreted as the model's understanding expressed in text.

## 3.1 Attention based Visual Token Selection

Several works have explored token selection based on attention weights [37, 13, 26]. These methods typically involve dropping tokens from early layers to enhance the efficiency of processing long contexts in later layers. Evidence from these studies suggests that a model's attention can serve as a strong indicator of the relevance of visual tokens.

Our goal is to globally select tokens from a short clip after it has been processed by the VideoLLM, providing context for subsequent clips. These tokens will guide the LLM's attention in the next short clip to focus on content that is coherent with past events, thereby reducing attention to background tokens, noise, and redundant visual tokens.

**Short Clip Processing.** We employ a pre-trained video-LLM, which consists of a visual encoder VE, an LLM and a projector module P. Given a $T$-frame video clip $\mathbf{V}$ and an instruction $\mathbf{X_I}$, outputs a textual caption of the video, $C$:

$$\mathbf{X}_V = \mathrm{P}(\mathrm{VE}(\mathbf{V})), \quad C = \mathrm{LLM}(\mathbf{X}_V, \mathbf{X}_I) \tag{1}$$

where $\mathbf{X}_V \in \mathbb{R}^{TN_V \times D}$ and $N_V$ denotes the number of spatial tokens per frame, and $D$ is the dimension of the LLM tokens. The target is to select a sparse set of tokens $S \in \mathbb{R}^{T \times N_S \times D}$ where $N_S \ll N_V$.

**Attention Coefficients Calculation.** Let us denote $\mathbf{X}_C$ the embedding of the generated caption. By the last generated token, attention coefficient tensor is estimated between the generated text tokens and the input tokens. We define the attention matrix $A^{l,h} \in \mathbb{R}^{(TN_V+N_I+N_C) \times (TN_V+N_I+N_C)}$ calculated on $\mathbf{X}^l = [\mathbf{X}_V^l, \mathbf{X}_I^l, \mathbf{X}_C^l]$ in a given layer $l$ and a given attention head $h$ where $N_I$ is the number of text tokens in the instruction prompt and $N_C$ is the number of tokens in the number of generated caption tokens.

After transferring $\mathbf{X}^l$ into Key $\mathbf{K}^{l,h}$, Value $\mathbf{V}^{l,h}$ and Query $\mathbf{Q}^{l,h}$, the attention scores $\mathbf{A}^{l,h}$ are computed using the scaled dot-product attention:

$$\mathbf{Q}^{l,h} = \mathbf{X}^l \mathbf{W}_Q^{l,h}, \quad \mathbf{K}^{l,h} = \mathbf{X}^l \mathbf{W}_K^{l,h} \quad \mathbf{V}^{l,h} = \mathbf{X}^l \mathbf{W}_V^{l,h} \tag{2}$$

where $\mathbf{W}_Q^{l,h}, \mathbf{W}_K^{l,h}, \mathbf{W}_V^{l,h}$ are the learnable weight matrices for queries, keys, and values, respectively[1].

$$\mathbf{A}^{l,h} = \mathrm{Softmax}\left(\frac{\mathbf{Q}^{l,h}(\mathbf{K}^{l,h})^\top}{\sqrt{d_k}}\right) \tag{3}$$

where $d_k$ is the dimension of attention heads.

First we extract the attention coefficients representing the cross attention between the generated caption and the input visual tokens:

$$\mathbf{A}_V^{l,h} = \mathbf{A}^{l,h}[TN_V + N_I : TN_V + N_I + N_C, \, 0 : TN_V] \tag{4}$$

---

[1]Our method focuses on attention weights, making operations involving the values $\mathbf{V}^{l,h}$ unnecessary and thus discarded.

We then calculate an attention score for each input visual token $\mathbf{X}_{Vj,:}$ by averaging the attention scores from all generated caption tokens $N_C$. Finally, the global importance score for a given visual token $\mathbf{X}_{Vj,:}$ is determined by averaging the attention scores across different attention heads $H$ and layers $L$, representing the overall attention of the model.

$$a_j = \frac{1}{L}\sum_{l=1}^{L}\frac{1}{H}\sum_{h=1}^{H}\left(\frac{1}{NC}\sum_{i=1}^{N_C}\mathbf{A}_V^{l,h}{}_{ij}\right) \tag{5}$$

In practice, to limit the compute cost, we consider only a subset of layers. Note that instead of averaging considering other operations such as max pooling is straight forward.

**Selecting Top Tokens:** Finally, the top $N_S$ tokens of $\mathbf{X}_V$ with the highest attention coefficients are selected: $\qquad\qquad \mathbf{S} = \mathbf{X}_V[\pi(1),\pi(2),\ldots,\pi(N_S),:] \tag{6}$
where $\pi = \mathrm{argsort}(\mathbf{a})$ are the indices that sort $\mathbf{a}$ in descending order [2].

## 3.2 Long Video Recurrent Processing.

For the very first short clip $\mathbf{X}_V^{(0)}$ we have selected a set of $N_S$ tokens $\mathbf{S}^{(0)}$ receiving the highest attention from the generated caption. Now for a next shortclip $\mathbf{X}_V^{(1)}$, we provide as input to the LLM the previously selected tokens $\mathbf{S}^{(0)}$ as input along with the full set of tokens comprising the current video clip $\mathbf{X}_V^{(1)}$. Similarly, after generating a caption $C^{(1)}$ we select a sparse set of most relevant tokens receiving highest attention scores from $\mathbf{X}_V^{(1)}$: $\mathbf{S}^{(1)}$.

For the following video shortclips, we create a FIFO queue of past selected tokens $[\mathbf{S}^{(0)},\mathbf{S}^{(1)},\ldots,\mathbf{S}^{(t)}]$ which are provided to the LLM as context a long with the next short clip raw tokens $\mathbf{X}_V^{(t+1)}$ up to the limit of the context window length $W$ or compute constraint.

## 3.3 Efficient Video Question Answering

We have illustrated how to process each video clip tokens $\mathbf{X}_V^{(t)}$ representing $T$ frames while having access to a long window of past selected tokens. During this process a caption $C^{(t)}$ is generated, capturing the LLM thoughts and understanding about the current clip events conditioned on past clips provided in the context window. We store these generated captions embeddings $\{\mathbf{X}_C^{(t)}\}$ and given a question $q$, we compute the average cosine similarity between the question tokens $\mathbf{X}_q$ and the tokens of each stored caption $\{\mathbf{X}_C^{(t)}\}$. Instead of retrieving the top $K$ captions based solely on average cosine similarity to the query $q$, we use Maximal Marginal Relevance (MMR) [4], a common retrieval technique where a score is calculated by balancing relevance to the query (cosine similarity between caption and query) with diversity among the selected captions (cosine similarity between candidate caption and already selected ones). Utilizing this technique, we effectively reduce potential redundancy that can emerge in

**Algorithm 1** Streaming Video Processing and Query Answering with rLiVS

**Streaming Video Processing**
1: $M_l \leftarrow [\,]$, $\quad M_s \leftarrow$ queue(), $\quad B \leftarrow [\,]$
2: MAX_MEM $\leftarrow$ 16, $\quad$ CLIP_SIZE $\leftarrow$ 16
3: **while** frames available **do**
4: $\quad$ $B$.append(get_next_frame())
5: $\quad$ **if** length($B$) == CLIP_SIZE **then**
6: $\quad\quad$ context $\leftarrow M_s$ + $B$
7: $\quad\quad$ $B$.clear()
8: $\quad\quad$ $S, C \leftarrow$ Attn_Selection(context)
9: $\quad\quad$ **if** length($M_s$) == MAX_MEM **then**
10: $\quad\quad\quad$ $M_s$.pop_left()
11: $\quad\quad$ **end if**
12: $\quad\quad$ $M_s$.append($S$)
13: $\quad\quad$ $M_l$.append($C$)
14: $\quad$ **end if**
15: **end while**
**Query Answering**
16: $Q \leftarrow$ embed(query)
17: $C' \leftarrow$ Retrieve_TopK($Q, M_l$)
18: context $\leftarrow C' + Q$
19: answer $\leftarrow$ LLM_Generate_Answer(context)

recurrent caption generation, while promoting coverage of distinct information when answering. Algorithm 1 summarizes our approach.

Our simple design and reliance on generated captions for video question answering is stemmed from the efficiency of storing the caption tokens IDs and the utility of similarity estimation between textual

---

[2]Basically, we select the top $N_S$ visual tokens (based on the attention scores), and store them in their original temporal order.

tokens without the reliance on any external embedding encoder such as CLIP [25]. Although we can store and provide both visual tokens $\mathbf{X}_V^{(t)}$ and captions $\mathbf{X}_C^{(t)}$ to the LLM for question answering, our experiments show that video-LLMs perform better with only caption input.

## 4 Experiments

In this section, we evaluate our proposed method against state-of-the-art approaches in streaming and long video understanding benchmarks, demonstrating strong performance and notable efficiency across experiments.

**Evaluation Benchmarks.** We evaluate our method's effectiveness in online scenarios using the Realtime VStream-QA benchmark [38], which includes *RVS-Movie* (emphasizing plot understanding) and *RVS-Ego* (focusing on visual comprehension)—both featuring 40-minute videos with diverse open-ended questions. To demonstrate robustness, we also report results on offline benchmarks: *MovieChat* [28] (170 videos averaging 576 seconds across various genres with 510 questions testing long-range comprehension), offline VStream-QA (*VS-Movie* and *VS-Ego*), and *CG-Bench* [5] (1,219 videos averaging 27 minutes with 12K multiple-choice questions). Additionally, we conduct an ablation on *NextQA*-valset [36] (570 shorter videos averaging 44 seconds with 5K multiple-choice questions) to validate our attention-based visual token selection approach.

**Baselines.** We compare our method against established video LLMs and recent long video understanding approaches. Training-required baselines include: VideoChatGPT [22] (employing temporal-spatial pooling), Chat-UniVi [14] (offering unified image/video representations through a set of dynamic visual tokens), LLaMA-VID [21] (using learning-based compression with context/content tokens per frame), VideoScan [20] (compressing frames into single learned semantic tokens preserving temporal context), and Flash-VStream-7B [38] (processing frame-by-frame with hierarchical memory including FIFO spatial memory, weighted k-means temporal consolidation, and past-current information abstraction). All operate on non-overlapping sliding windows. Training-free baselines include: MovieChat [28] (using FIFO short-term memory and similarity-based long-term consolidation), Goldfish [1] (processing independent short clips with caption generation and retrieval), and ReKV [9] (storing full KV-Cache retrieved through cross-attention to queries).We coin our method rLiVS as a short for recurrent LLM-informed Visual Selection.

**Evaluation Metrics.** For benchmarks with open-ended questions, we report both Accuracy and a Score based on GPT-3.5 evaluation, following prior works (we use `gpt-3.5-turbo-0613`, consistent with the evaluation setup of ReKV [9]). The model is given the question, ground truth, and prediction, and asked to assess whether the prediction matches the answer ("Yes/No") and to assign a compatibility score. The prompt template is provided in the Appendix. For multiple-choice benchmarks, we compute accuracy by directly comparing predictions to ground truths. Additionally, for the streaming benchmark, we report end-to-end latency, GPU usage (from query arrival to response), and KV-Cache utilization. Benchmarkings are conducted on a single A100 GPU.

**Implementation Details.** We implement our method on LLaVA-OneVision [17], a strong VLM for image and video tasks, enabling direct comparison with ReKV (current state-of-the-art on RVS benchmarks). We demonstrate versatility by evaluating both 7B and 0.5B variants, with 7B used unless specified otherwise. Since LLaVA-OV is trained with 32 frames (196 visual tokens each), we allocate 16 frames for current short clips and 16 for short-term recurrent memory. We select only 196 visual tokens from 3,136 available (16×196), retaining just 6.25% of total visual information per short clip. Following previous works, we process RVS-Movie and RVS-Ego at 0.5 FPS [38, 9], MovieChat at 1 FPS, and CG-Bench and offline VS-Stream at 0.5 FPS, with 10K context tokens for retrieval and generation. We average attention scores from 4 (of 28) backbone layers across experiments. To show generalization across video-LLM backbones of our streaming video pipeline, we also incorporate rLIVS to Qwen2.5-VL-7B [3] and test on RVS streaming benchmark. We provide details for this setup in Appendix A. Finally, in the Appendix we include ablations on layer selection (Appendix B), as well as on attention aggregation methods (average vs. max), and analysis of how context length affects performance and latency in Appendices C and D, respectively.

## 4.1 Results

We next present results across the discussed benchmarks: starting with an ablation on visual selection methods for short video understanding, followed by evaluations on offline long video QA, and concluding with streaming video understanding.

Table 1: Comparison on offline long video benchmarks. Accuracy (Acc.) and Score (Sco.) are reported where applicable. Results for ReKV are taken from [9], for VS-Stream from [38], for Moviechat from [28] and [1], while for CG-Bench from [5].

| Method | VS-Ego | | VS-Movie | | MovieChat | | CG-Bench |
|---|---|---|---|---|---|---|---|
| | Acc. | Sco. | Acc. | Sco. | Acc. | Sco. | Acc. |
| Video-ChatGPT [22] | 51.7 | 3.7 | 54.4 | 3.4 | 47.6 | 2.5 | - |
| MovieChat [28] | 52.2 | 3.4 | 39.1 | 2.3 | 62.3 | 3.2 | - |
| Chat-UniVi [14] | 50.9 | 3.8 | 54.0 | 3.4 | - | - | 25.9 |
| LLaMA-VID [21] | 54.8 | 3.9 | 51.4 | 3.4 | 53.2 | 3.8 | - |
| Goldfish [1] | - | - | - | - | 67.6 | 4.2 | - |
| Flash-VStream-7B [38] | 59.0 | 3.9 | 56.1 | 3.4 | - | - | - |
| **rLiVS (Ours)** | **61.0** | 3.9 | **59.3** | 3.6 | **78.0** | 4.0 | **33.1** |

### 4.1.1 Evaluation of Visual Token Selection Strategies on Short Video Datasets

Visual token selection is central to our method, as it brings past information into current short-clip processing via recurrency, enhancing captioning and understanding of the overall video stream. We argue that naive methods such as uniform sampling or mean pooling risk discarding important information, while traditional clustering approaches like K-Means can be slow and sub-optimal in high-dimensional spaces. Instead, by leveraging the LLM's attention—already computed during caption generation—we gain a powerful, low-overhead signal that enables aggressive compression with minimal performance loss. To empirically validate this, we conduct experiments on the NextQA-valset, a short video benchmark, to assess the token's selection quality independently from the factors of long video understanding. We compare the full model's performance while retaining only 6% or 12% of visual tokens—selected with different methods:

Table 2: Accuracy (Acc.) of different selection methods evaluated on NextQA-valset.

| Selection Method | Next-QA (valset) |
|---|---|
| | Acc. |
| Full Model | 78.6 |
| Uniform Sampling (6%) | 75.5 |
| Mean Pooling (6%) | 70.7 |
| K-Means (6%) | 76.8 |
| Attention (ours) (6%) | **77.0** |
| Uniform Sampling (12%) | 76.7 |
| Mean Pooling (12%) | 75.5 |
| K-Means (12%) | 78.1 |
| Attention (ours) (12%) | **78.4** |

1) via uniform sampling, 2) via mean pooling, 3) via K-Means and 4) based on our attention selection mechanism to the generated caption, as described in Section 3.1. Table 2 confirms our intuition; while uniform sampling results in 2-3% performance loss, our attention-based selection outperforms uniform sampling even at twice the compression rate—incurring only a 1.5% performance drop with 6% of visual tokens, and nearly matching full-model performance at 12%. Moreover, mean pooling, naively combining visual tokens without considering their relevance, yields the worst performance, while K-Means performs comparably to attention-based selection but it is significantly slower—especially when operating in the high-dimensional input space of the LLM—introducing unnecessary computational overhead.

### 4.1.2 Offline Video Question Answering Datasets

In Table 1 we present results on offline long video understanding benchmarks. rLiVS incorporated in LLaVA-OV-7B achieves state-of-the-art performance in VS-Ego and VS-Movie benchmarks, outperforming the previous best by 2% and 3%, respectively. On MovieChat and CG-Bench, our method achieves strong results, outperforming prior approaches, demonstrating the effectiveness of heavily compressing visual tokens without harming performance. Overall, these results indicate that our method serves as a strong baseline, efficiently handling long inputs, requiring no additional training or fine-tuning—while maintaining minimal memory storage requirements.

Table 3: Evaluation on the Realtime VStream-QA streaming benchmark [38], consisting of RVS-Ego and RVS-Movie subsets. Accuracy (Acc.), Score (Sco.), Latency, VRAM and KV-Cache usage are reported. Results for baselines are taken from [20].

| Method | RVS-Ego | | RVS-Movie | | Latency | VRAM | KV-Cache |
|---|---|---|---|---|---|---|---|
| | Acc. | Sco. | Acc. | Sco. | | | |
| MovieChat [28] | 50.7 | 3.4 | 36.0 | 2.3 | - | - | - |
| LLaMA-VID [21] | 53.4 | 3.9 | 48.6 | 3.3 | - | - | - |
| Flash-VStream-7B [38] | 57.3 | 4.0 | 53.1 | 3.3 | 2.1s | 19GB | - |
| VideoScan [20] | 60.9 | 4.0 | 54.1 | 3.5 | 2.1s | 18GB | - |
| *LLaVA-OV 0.5B* | | | | | | | |
| ↪ ReKV [9] | 54.7 | 3.7 | 44.6 | 3.4 | 1.6s | 19GB | 4.0 GB/h |
| ↪ **rLiVS (Ours)** | 57.6 | 3.8 | 51.3 | 3.4 | 1.5s | 11GB | - |
| *LLaVA-OV 7B* | | | | | | | |
| ↪ ReKV [9] | 63.7 | 4.0 | 54.4 | 3.6 | 2.7s | 36GB | 18.8 GB/h |
| ↪ **rLiVS (Ours)** | 65.3 | 4.0 | **57.7** | 3.6 | **1.9s** | 25GB | - |
| *Qwen2.5-VL 7B* | | | | | | | |
| ↪ **rLiVS (Ours)** | **68.1** | 4.0 | 56.1 | 3.6 | 2.7s | 19GB | - |

#### 4.1.3 Streaming Video Question Answering Datasets

Finally, we present our results on RVS-Ego and RVS-Movie, long video streaming benchmarks that represent the primary focus of this work and the main setting for evaluating our proposed method. rLiVS demonstrates strong suitability for this setting as shown in Table 3, outperforming the previous best, ReKV, with the same backbone model, by 2–3% and *achieving new state-of-the-art results on the benchmark, while also being nearly 1 second faster than ReKV*. Importantly, our attention-based compression technique—combined with recurrency and caption-guided retrieval and answering—proves robust across both subsets of the streaming benchmark. Unlike ReKV, rLiVS does not require external memory offloading and operates within a 10K context window, enabling both efficient and effective performance, while resulting in 11GB less peak VRAM usage with the same video-LLM backbone. Interestingly, when paired with the lightweight 0.5B model, our method outperforms the previous second-best approach that uses a 7B model, as well as ReKV with the same 0.5B model by 2.9% on RVS-Ego and 6.7% on RVS-Movie, highlighting the strength of caption-based answering even in smaller model variants emphasizing the efficient aspect of our design choices. Finally, by coupling our method with the recent strong multimodal LLM Qwen2.5-VL, we further improve performance on RVS-Ego, reaching 68.1% accuracy. *This highlights the model-agnostic and plug-and-play nature of our porposed rLiVS*.

A key strength of our framework is its ability to adapt token selection dynamically based on task instructions, which serve as top-down signals. Through instruction-guided attention, the model selects a sparse subset of semantically relevant visual tokens, enabling efficient, context-aware understanding. As shown in Appendix H, token selection is highly sensitive to the instruction, with different tasks yielding distinct attention patterns. This adaptivity supports a broad range of video understanding tasks without retraining or architectural changes, highlighting promising directions for scalable and generalizable long video models.

#### 4.1.4 Ablation of rLiVS Design Choices

Next, we ablate our method's core design choices through targeted experiments, demonstrating the following: (1) the importance of recurrency when processing long videos split in short clips; (2) the superiority of captions over selected visual tokens—or their combination—for retrieval and answering in long video streams, (3) the performance benefits of LLM-

Table 4: Ablation on the effect of accessing past visual information via recurrency and attention selection.

| Method | RVS-Ego | | RVS-Movie | | MovieChat | |
|---|---|---|---|---|---|---|
| | Acc. | Sco. | Acc. | Sco. | Acc. | Sco. |
| rLiVS | **65.3** | **4.0** | **57.7** | **3.6** | **78.0** | **4.0** |
| *w/o recurrency* | 62.5 | 3.9 | 53.7 | 3.5 | 74.1 | 3.9 |

informed attention-based selection compared to naive uniform sampling in streaming scenarios; and (4) the effect of compression rate on downstream performance.

**The importance of recurrency in long streams.** As previously discussed, [1] proposes processing long videos as independent short clips, storing only the textual outputs (captions) generated by a video-LLM and retrieving relevant information based on cosine similarity to the input question. While this approach is simple and efficient, it suffers from a lack of continuity that becomes increasingly problematic as video length increases. Specifically, the absence of shared visual information across clips and sole reliance on textual representations can hinder the model's ability to consistently track entities—such as people or objects—across time, especially when captions lack distinguishing context. These limitations naturally impact performance. To empirically prove this, we present results (Table 4) on RVS-Ego, RVS-Movie, and MovieChat, showing that introducing recurrency consistently improves performance by 3–4%, under otherwise identical conditions. Importantly, recurrency in our method serves a dual purpose: (1) it enhances continuity and coherence across short clips, boosting long video understanding, and (2) it guides the LLM's attention during visual token selection, reinforcing its own impact.

**Captions vs Visual Tokens for *retrieval* and *answering*.** A natural question is whether textual information (captions) or selected visual tokens provide a better signal for retrieving and answering user queries—that is, which representation is the best choice for effectively preserving crucial information while enabling compression. **For retrieval**, the input query is embedded into the LLM's input space. Visual tokens, projected ad hoc from the visual encoder, lack learned semantic alignment with queries, leading to less meaningful similarity scores (as illustrated in Appendix F). Proper similarity between them would require projecting both queries and visual tokens into a common space (like CLIP encoders [25]), adding extra computational steps to our pipeline.

In contrast, captions share the same modality as queries and can be directly embedded to the LLM's input space, making them naturally better suited for retrieval via simple cosine similarity. **For answering**, Table 5 presents results using visual tokens, captions, or their combination as retrieved information for each short clip. Among the approaches, using only captions achieves the best performance. Despite visual tokens theoretically offering richer information, they under-perform likely due to the

Table 5: Comparison of vision vs. language modalities for information retrieval and query response on RVS-Ego/Movie benchmarks.

| Retrieved Modality | RVS-Ego | | RVS-Movie | |
|---|---|---|---|---|
| | Acc. | Sco. | Acc. | Sco. |
| Selected Visual Tokens | 58.2 | 3.9 | 48.4 | 3.5 |
| Captions | **65.1** | 4.0 | **57.7** | 3.6 |
| Combination | 63.0 | 4.0 | 54.3 | 3.5 |

mismatch between video-LLM training data (predominantly short clips of a few seconds) and evaluation contexts (minutes/hour steaming videos). Current video-LLMs struggle to comprehend visual information at larger timescales, even when computationally representable as input. This aligns with findings from [33], which demonstrated that despite the existence of architectural optimizations that enable long video processing (through memory structures or compressed representations), video-LLMs still demonstrate significant performance degradation in these contexts. In contrast, LLM research has extensively explored long-context adaptation, developing effective modeling and training techniques that enable reasoning over extended sequences of textual inputs, such as multi-clip captions. Thus by utilizing only the captions as past information during retrieval and answering, we effectively transform the problem into a text-based long context QA problem, harnessing LLMs' proven strength in reasoning over these extended contexts.

**Attention-based vs Uniform Visual Selection on streaming scenarios.** We previously demonstrated (Sec. 4.1.1) that our attention-based selection outperforms uniform sampling in short video QA, even at twice the compression rate, enabling aggressive token reduction with minimal performance loss. We now extend this evaluation to the streaming setting, where selection drives captioning (through recurrency) and thus plays a critical role in overall performance. Our results in Table 6 are consistent with those

Table 6: Comparison of selection methods within our pipeline on RVS-Ego and RVS-Movie.

| Selection Method | RVS-Ego | | RVS-Movie | |
|---|---|---|---|---|
| | Acc. | Sco. | Acc. | Sco. |
| Uniform Sampling | 64.2 | 3.9 | 56.0 | 3.5 |
| Attention (ours) | **65.1** | 4.0 | **57.7** | 3.6 |

in the short clip benchmark: attention-based selection, leveraging LLM's ability to attend to crucial information, effectively filtering out noise and redundancy, is superior in both subsets by 1-2%, without introducing any additional overhead, since caption generation is part of our pipeline. Note that uniform sampling is integrated into our full pipeline, replacing only the selection strategy. This approach serves as a significantly stronger baseline compared to the uniform sampling baseline of ReKV [9], as demonstrated in the Appendix.

**Compression Rate and Downstream Performance.**
Our choice of aggressive visual token compression (e.g., retaining only 6%) may raise concerns about potential information loss and downstream performance degradation. However, this reflects a practical trade-off in real-time video understanding. Long-form videos often contain substantial visual redundancy, allowing for effective token filtering without harming performance. To validate this, Table 7 shows how accuracy varies with different compression rates on the NextQA validation set. Retaining 12% of visual tokens nearly matches full performance, while 6% leads to negligible loss[3]. Our framework remains flexible, with the compression ratio tunable to task requirements.

Table 7: Acc. on NextQA valset for different % of visual tokens selected with our proposed attention-based method.

| Selected Tokens | Acc.(%) |
|---|---|
| 1% | 68.0 |
| 6% | 77.0 |
| 12% | 78.4 |
| 19% | 78.7 |
| 25% | 79.0 |
| 100% | 78.6 |

Looking ahead, adaptive compression—dynamically adjusting retention based on visual content complexity—is a promising direction. While it may improve performance in visually rich scenes, it introduces computational overhead, which we currently avoid to maintain online efficiency. Qualitative results in Appendix J further demonstrate the robustness of our method under extreme compression.

## 5 Discussion

Streaming video question answering and understanding is an emerging landscape that can enable the reliable deployment of intelligent agents in various domains requiring visual comprehension. A key design requirement for practical solutions is low memory and compute usage. In this work, we propose a simple approach based on two design choices: first, the instant selection of visual tokens based on the LLM's attention, resulting in $\sim 95\%$ compression rate of short clip content; second, the use of recurrent processing and caption based question answering to smoothly enable higher-level reasoning over longer time spans. With these components, we achieve state-of-the-art performance on streaming video benchmarks. Our approach is agnostic to the video-LLM deployed and requires no training, setting a strong baseline for future work.

Beyond question answering, our framework shows potential as a general-purpose backbone for long-form video processing and understanding tasks such as summarization, retrieval, and object tracking. Preliminary results on long video summarization (Appendix I) provide initial evidence of the method's generalizability. Additionally, our analysis in Appendix H demonstrates that the memory selection mechanism is both task-sensitive and instruction-conditioned, underscoring its flexibility and adaptability. We view this as a promising direction for future research, where natural language task descriptions can dynamically guide memory selection and reasoning.

**Limitations.** Despite strong performance, our method prioritizes efficiency over completeness by attending only to selected content during short clip processing. This may overlook fine-grained details and harm continuity, as the LLM lacks access to the full video history. While our FIFO memory buffer offers short-term context and captions serve as long-term memory, this temporal mechanism may not always capture semantically salient information. A promising direction is to explore more semantic, rather than purely temporal, memory selection strategies that remain efficient.

Our approach is also training-free and fully reliant on pre-trained backbones, inheriting their limitations in visual understanding and temporal reasoning. The recurrent captioning process, conditioned on prior frames and instructions, can introduce redundancy across captions, which may affect retrieval performance. We partially address this by considering both novelty and relevance during retrieval. Future work includes integrating the method into training pipelines and evaluating across a wider range of models and benchmarks to better understand its strengths and limitations.

---

[3]An additional benefit is that selecting only 6% of tokens allows more memory slots in the FIFO queue, enhancing long-term understanding.

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

# Technical Appendices and Supplementary Material

## A   Experimental Details

In this section, we provide a detailed description of our evaluation setup and methodological choices to ensure the reproducibility of our experiments:

- *Visual Tokens Selection*: As described in the main paper, our attention-based selection method selects $k$ out of $N$ visual tokens globally. The backbone video-LLM we use (LLaVA-OV) is trained on 32 video frames, each contributing 196 tokens, resulting in a total of 6,272 visual tokens. We split the 6,272-token context window into two halves: the first is used to encode past information (by injecting previously selected visual tokens), and the second is reserved for the raw visual tokens of the current short clip. We choose $k$ such that it evenly divides the 3,136 tokens of the second half (current short clip) in order to fully utilize the context. During selection, we consider only on the second half, corresponding to the current short clip (i.e., 3,136 tokens). For both long-video offline and streaming benchmarks, we select 196 tokens from this set, which is $\frac{1}{16}$ (or approximately $6.25\%$) of the total. For simplicity, we report this as $6\%$s throughout the paper. In our uniform baseline, everything remains the same and we just sample 196 out of the 3,136 tokens of the current short clip.

- *Resources*: Across all experiments, we utilize one or two NVIDIA A100 GPUs with 40GB of memory. Latency and VRAM measurements are conducted on a single GPU.

- *LLM Evaluation*: As reported in the main paper, we follow prior work [9, 38] for LLM-based evaluation and use `gpt-3.5-turbo`. The prompt templates employed are provided in Section K.

- *Qwen2.5-VL setup*: This model dynamically embeds frames based on input resolution, resulting in different number of tokens per frame. In our experiments, we use the same number of tokens per short clip as LLaVA-OV to ensure fair comparison.

## B   Layer Selection for Attention-Based Visual Token Selection

In our method, visual tokens are selected based on their attention scores to the generated captions. Rather than using attention from all layers, we compute scores from a small subset. Modern fast attention implementations (e.g., FlashAttention-2 [8]) avoid explicitly materializing the full $N^2$ attention matrix (with $N$ as the sequence length) to reduce memory and computational cost. To maintain efficiency and avoid additional overhead, we use attention from only a small $\ell$ out of $L$ total number of layers (28 in LLaVA-OV). In the main paper experiments, we set $l = 4$ and we select these layers via uniform sampling across the depth of the network. As shown in Table 8, this simple strategy provides robust performance comparable to using all layers, with minimal variance. In contrast, selecting layers from localized regions (e.g., early, middle, or late) yields less stable results, and using a single layer results in significantly worse performance.

## C   Methods for Aggregating Attention Scores in Token Selection

In our method, we opted for average score over different attention heads for token selection; however, we noted that alternative score aggregation strategies could also be explored. Here we present results on the NextQA-valset using *max* as the aggregation method and selecting $6\%$ of the visual tokens (to directly compare with the results of the main paper). As shown in Table 9, both methods perform similarly, with a negligible difference of only $0.1\%$ in accuracy.

## D   Context Length for Retrieving & Answering

In the main paper, we use a context length of 10k tokens for retrieving and answering user queries. This choice is motivated by two key factors: (1) **Latency**: In streaming video applications, where user

---

[4]These layers are used across experiments in the main paper.

Table 8: Accuracy on NextQA validation set for different layer selections, keeping 6% of visual tokens.

| Layers | Acc. on NextQA-valset (%) |
|---|---|
| Full Model | 78.6 |
| *all layers* | 76.8 |
| *[5, 9, 14, 20]*[4] | 77 |
| *[3, 12, 18, 24]* | 76.8 |
| *[1, 10, 15, 25]* | 77.2 |
| *[4, 5, 16, 28]* | 76.7 |
| *[3, 4, 5, 6]* | 75.2 |
| *[13, 14, 15, 16]* | 76.6 |
| *[24, 25, 26, 27]* | 76.6 |
| *[3]* | 73.3 |
| *[15]* | 76.4 |
| *[26]* | 75.7 |

Table 9: Accuracy on NextQA validation set for different layer aggregation methods of attention heads, keeping 6% of visual tokens.

| Aggregation Method | Acc. on NextQA-valset (%) |
|---|---|
| Full Model | 78.6 |
| Avg. | 77.0 |
| Max | 77.1 |

queries arrive alongside frames in real-time, low-latency responses are critical. Increasing the context length substantially slows down both retrieval and LLM inference. (2) **Noise Filtering**: Long video streams are highly redundant, and larger contexts increase the risk of including irrelevant information, which can degrade performance. We empirically validate these considerations in Table 10 on RVS-Movie and RVS-Ego streaming benchmarks, showing that while increasing context from 6k to 10k improves accuracy, a further increase to 20k offers no additional benefit—and in some cases, even hurts performance—while incurring higher computational costs. Overall, a 10k context length offers a favorable trade-off between effectiveness and efficiency. Finally, we note that ReKV [9] retrieves 64 frames (each consisting of 196 tokens, since the same model is used), resulting in a larger context and, thus, in higher latency (2.7s) as reported in the main paper.

Table 10: Comparison of accuracy and score under different context lengths on RVS-Movie and RVS-Ego.

| Context Length | RVS-Ego | | RVS-Movie | | Latency |
|---|---|---|---|---|---|
| | Acc. | Score | Acc. | Score | |
| **20k** | 65.3 | 4.0 | 56.0 | 3.5 | 3.9s |
| **10k** | 65.3 | 4.0 | 57.7 | 3.6 | 1.9s |
| **6k** | 62.7 | 3.9 | 57.0 | 3.5 | 1.6s |

# E  Positional Information

A potential concern with our proposed method is the manipulation or loss of spatial positional information during visual token selection. In rLiVS, we retain only the temporal order of selected tokens, potentially altering their original spatial layout when reinserting them into the LLM context. However, our initial experiments suggest that this design choice does not hinder performance. As shown in Table 7, selecting only 19% of the visual tokens—based on our relevance scoring and preserving temporal order—achieves comparable performance to using the full set of visual tokens on the NextQA validation set. Interestingly, selecting 25% of tokens even outperforms the full-token baseline, indicating that reducing redundancy and filtering out less informative content is more beneficial than strictly preserving spatial structure. This finding aligns with prior observations (e.g.,

[13]) that video data contains substantial redundancy, which can degrade performance if not mitigated. Additionally, since vision-language models (VLMs) are primarily pre-trained for caption generation, we posit that leveraging this objective as an intermediate step acts as an effective noise filter. During pretraining, these models learn to focus on the most informative visual tokens—those that align with the semantics of captions. Our method builds directly on this property by selecting tokens that contribute most to caption generation, thereby enhancing relevance and reducing noise. Finally, we note that by choosing not to address the challenge of accurately injecting positional information, our method remains orthogonal to any specific video-LLM architecture. Developing improved strategies to explicitly retain or reintroduce spatial information—and thereby better model spatio-temporal dynamics during processing—remains an open research direction, which we leave for future work.

## F    Visual tokens vs Captions based Retrieval

In our ablation studies, we argue that visual tokens projected ad hoc from the visual encoder lack learned semantic alignment with user questions. As a result, they produce uninformative similarity scores and lead to confusing retrieval behavior. In contrast, captions—being in the same modality as user queries—naturally lie within the same LLM input space, making them more suitable for retrieval. Figure 2 illustrates the distribution of cosine similarity scores between visual tokens and queries, as well as between captions and queries, aggregated across videos and questions in RVS-Movie. We observe that the similarity distribution between visual tokens and queries is narrowly centered around zero, with values ranging from approximately -0.02 to 0.06. In contrast, the similarity scores between captions and queries span a much broader range—from about $0.4$ to $0.9$—nearly an order of magnitude wider—potentially indicating a greater capacity to distinguish relevant from irrelevant content during retrieval.

To verify this point, we present performance results on both RVS-Ego and RVS-Movie using three similarity-based retrieval methods: visual tokens, captions, and random selection[5]. As shown in Table 11, retrieval using visual tokens performs on par with random selection, reinforcing the claim that visual tokens provide limited value. In contrast, caption-based retrieval consistently yields improvements of approximately 2–3%, underscoring its effectiveness.

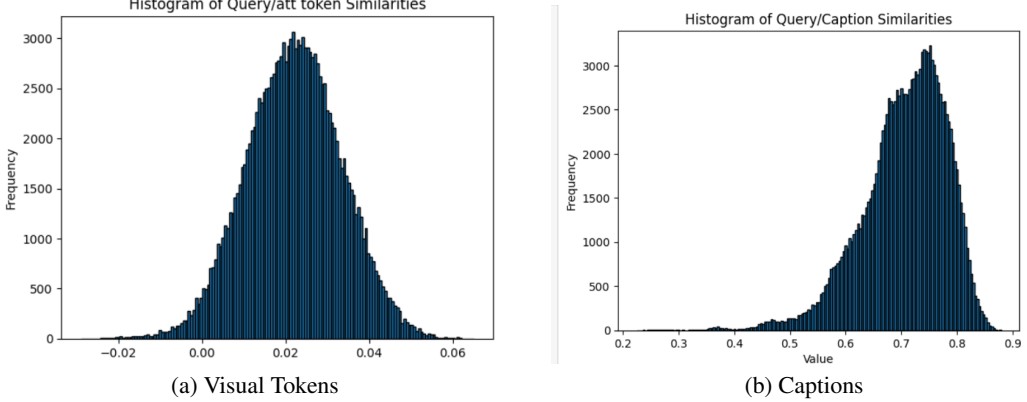

|          (a) Visual Tokens          |          (b) Captions          |

Figure 2: Distribution of cosine similarities between visual tokens and queries (left), and between captions and queries (right), averaged over videos and questions in RVS-Movie.

## G    Our uniform baseline vs ReKV's [9]

In the main paper, we compared attention-based and uniform visual token selection on the streaming benchmarks RVS-Ego and RVS-Movie, showing that our attention-based method consistently outperforms uniform sampling by approximately 1–2%. Notably, the uniform baseline in that comparison was strengthened with two key components of our approach: recurrency and short-clip level token selection—both of which significantly improve performance. Here, we present a simpler uniform

---

[5]Across methods, captions are retrieved for answering.

Table 11: Comparison of accuracy and score under different retrieval strategies (based on captions, visual tokens or random) on RVS-Movie and RVS-Ego.

| Retrieval | RVS-Ego | | RVS-Movie | |
|---|---|---|---|---|
| | Acc. | Score | Acc. | Score |
| **Random** | 64.6 | 3.9 | 55.2 | 3.5 |
| **Visual Tokens** | 63.7 | 3.9 | 55.0 | 3.5 |
| **Captions** | 65.3 | 4.0 | 57.7 | 3.6 |

sampling baseline more aligned with the setting of [9], where tokens are sampled uniformly across the entire video. To adapt this to our framework, we uniformly select 32 frames globally to fill the context, consistent with the frame length that our video-LLM backbone was trained. As shown in Table 12, this naive baseline performs significantly worse—about 10% lower—than our strengthened uniform variant, highlighting the importance of our proposed design choices.

Table 12: Comparison of accuracy and score of global uniform sampling (sampling frames) vs our recurrent short-clip caption-based baseline on RVS-Movie and RVS-Ego.

| Baseline | RVS-Ego | | RVS-Movie | |
|---|---|---|---|---|
| | Acc. | Score | Acc. | Score |
| **Global Uniform** | 56.2 | 3.7 | 44.1 | 3.1 |
| **Recurrent short-clip caption-based Uniform** | 64.2 | 3.9 | 56.0 | 3.5 |

## H   Attention-based Selection based on Input Instruction

### H.1   Core Mechanism

We propose a flexible top-down attention mechanism in which input instructions guide the selection of visual tokens through a caption- or response-generation step. The overall pipeline is structured as follows:

*Instruction → Generated Caption/Response → Attention-Based Token Selection*

Tokens are selected based on their attention scores during text generation, aligning with task-specific semantic relevance. As discussed, this mechanism draws inspiration from human memory systems, wherein attention is modulated by task-driven goals rather than uniform processing across all inputs.

While prior work has leveraged attention scores for token reduction [13, 26, 37], these approaches typically operate within individual layers, pruning tokens based on localized attention during early processing stages. In contrast, our method introduces several key innovations. First, we compute *cross-layer attention aggregation*, identifying tokens that receive consistently high attention across multiple layers. Second, we apply *sparse token retention*, keeping only a small subset of visual tokens (approximately 6%) based on the aggregated scores. Third, through *recurrent token reuse*, the selected tokens are propagated across successive short video segments, enabling temporal continuity without any model fine-tuning. This design not only reduces computational cost but also preserves semantic coherence across clips, facilitating efficient and context-aware video reasoning.

### H.2   Task Adaptability

Although our main experiments focus on caption-guided token selection for visual question answering (VQA), the proposed framework is inherently instruction-adaptive and capable of generalizing across a variety of video understanding tasks through modification of the input instruction. For example:

- **Person Tracking**: "Track the movement of the person in the red shirt throughout this video sequence."

- **Action Recognition**: "Identify and describe the specific athletic movements performed."
- **Object Detection**: "Locate and describe all vehicles appearing in this scene."

Each task-specific instruction elicits distinct model outputs, leading to unique attention patterns and token selections that reflect the semantic priorities of the task. To qualitatively assess this behavior, we conducted a comparative analysis of token selection overlaps under varying instructions.

**Case Study: Instruction Sensitivity**   We selected two representative videos and evaluated the overlap of selected visual tokens using a generic captioning prompt versus task-specific instructions:

- **Video 1**: A girl with sunglasses is the main foreground object.
  - *Instruction*: "Track the locations of the girl wearing the sunglasses in the video."
  - *Token Overlap with Generic Prompt*: 44%
- **Video 2**: Two people playing cards, with interest focused on the background.
  - *Instruction*: "Locate and describe the objects appearing in the background of the video."
  - *Token Overlap with Generic Prompt*: 8%

The generic prompt used in both cases was: "Describe what's happening in the video," consistent with the one used during main captioning experiments. The results demonstrate that token selection is highly sensitive to the given instruction, validating the dynamic adaptability of our attention-based mechanism.

# I   Long Video Summarization Task

In this section, we conduct a preliminary evaluation of our method's task generalizability through long video summarization. Using the MLVU validation split [39], we compute the holistic video summarization score and compare our approach to Video-XL [27], a recent training-based streaming video understanding model that also employs recurrent mechanisms. As shown in Table 13, rLiVS outperforms Video-XL in holistic summarization quality.

Table 13: Holistic video summarization scores on MLVU (validation).

| Method | Model Size | MLVU Summarization Score |
|---|---|---|
| Video-XL | 7B | 3.40 |
| rLiVS (Ours) | 7B | **3.65** |

These results demonstrate that rLiVS not only matches but outperforms specialized, trained architectures like Video-XL while remaining training-free and easily extensible. We include a more detailed breakdown of this comparison in the revised version of the paper.

# J   Qualitative Analysis

To assess the effectiveness of our aggressive token compression in complex and dynamic scenarios, we conducted a qualitative evaluation on a subset of videos from the SportsQA dataset [18]. Although full quantitative results are pending due to dataset access constraints, we compare caption outputs generated using our method (6% token retention) against those generated using all visual tokens.

Despite the significant reduction in token count, the model retains its ability to produce semantically rich, fine-grained descriptions, closely matching the full-token baseline. Below are selected examples from the *aerobic gymnastics* category:

- **Full**: "The video showcases a synchronized gymnastics routine performed by a group of athletes at the 10th European Championships for Aerobic Gymnastics."
  **6%**: "The video is about a gymnastics routine performed by the Italian team at the 10th European Championships."

- **Full**: "The video showcases a rhythmic gymnastics performance by three athletes."
  **6%**: "The video showcases a rhythmic gymnastics routine performed by three athletes."
- **Full**: "The video showcases a synchronized gymnastics routine performed by two athletes at the 5th Aerobic Gymnastics Asian Championships."
  **6%**: "The video is about two gymnasts performing a synchronized routine at the 5th Aerobic Gymnastics Asian Championships."
- **Full**: "The video is about a rhythmic gymnastics performance by two athletes at the 7th FIG Aerobics World Age Group Competitions in Incheon, Korea."
  **6%**: "The video showcases a rhythmic gymnastics routine performed by two athletes."

These examples indicate that our model maintains high semantic fidelity, even under aggressive compression, and generalizes well to fast-changing scenes with multiple actors and fine-grained actions. This suggests the viability of our fixed but efficient token selection approach in high-complexity domains such as sports.

## K  Prompt Templates

We show here the prompts we use for caption generation and for the LLM evaluation:

**Caption Generation.** `"Describe what's happening in the video."`

**LLM evaluation.** For this, we follow previous works [38, 9] and utilize the following commonly used template:

```
messages=[
{
"role": "system",
"content":
"You are an intelligent chatbot designed for evaluating the correctness of
generative outputs for question-answer pairs.  "
"Your task is to compare the predicted answer with the correct answer and
determine if they match meaningfully.  Here's how you can accomplish the
task:"
"---"
"##INSTRUCTIONS: "
"- Focus on the meaningful match between the predicted answer and the
correct answer.\n"
"- Consider synonyms or paraphrases as valid matches.\n"
"- Evaluate the correctness of the prediction compared to the answer."
},
{
"role": "user",
"content":
"Please evaluate the following video-based question-answer pair:\n\n"
f"Question: {question}\n"
f"Correct Answer: {answer}\n"
f"Predicted Answer: {pred}\n\n"
"Provide your evaluation only as a yes/no and score where the score is an
integer value between 0 and 5, with 5 indicating the highest meaningful
match.  "
"Please generate the response in the form of a Python dictionary string
with keys 'pred' and 'score', where value of 'pred' is a string of 'yes' or
'no' and value of 'score' is in INTEGER, not STRING."
"DO NOT PROVIDE ANY OTHER OUTPUT TEXT OR EXPLANATION. Only provide the
Python dictionary string.  "
"For example, your response should look like this: {'pred': 'yes',
'score': 4.8}."
}
]
```

## L  Broader Impact

The broader impact of our approach can be discussed from three main aspects: the environmental impact, accessibility, and ethical consideration. Our method significantly reduces computational costs and energy consumption, subsequently lowering the carbon footprint associated with long video understanding. Along with the efficiency, our approach makes streaming video question answering readily accessible to practitioners and researchers with limited resources. However, as in the case of most AI research, our approach can potentially be misused, and it is crucial to ensure a responsible application. As such, preventing its use in unauthorized surveillance or privacy violations is important. In general, developing comprehensive guidelines and regulations on the ethical use of large video language models is essential to mitigate these risks.

