# OpenReview forum: "Recurrent Attention-based Token Selection for Efficient Streaming Video-LLMs"
_NeurIPS.cc/2025/Conference — NeurIPS 2025 poster_

### Official Review · Reviewer_hCyV · 2025-06-12

**Clarity:** 2
**Significance:** 2
**Originality:** 2
**Rating:** 4
**Confidence:** 4

**Summary:**

The paper proposes the Recurrent Attention-based Token Selection, which reduces the computational cost and improve the video LLM's performance. The strategies (LLM-informed selection of visual tokens, Hierarchical selection of tokens, Caption-based question answering) are used in the method. Experimental results demonstrate the method's effectiveness.

**Questions:**

Please refer to the main text

**Ethical Concerns:**

["NO or VERY MINOR ethics concerns only"]

**Final Justification:**

Thanks for the rebuttal. My concerns have been partly solved. rLiVS is different from Video-XL as a training free model. And the efficiency has been reported. Despite the author has not shown the model's performance on grounding, I tend to increase my score.

**Limitations:**

I haven't find the limitation section in the main text.

**Quality:**

3

**Strengths And Weaknesses:**

Strengths:
1. The method demonstrates the promising results.

Weaknesses:
1. The novelty is somewhat limited. I think the method is similar to Video-XL[1]. The difference is very minor (Both the two methods use the recurrent form to process the long video. The only difference is that the Video-XL uses the accumulated tokens, while the rLiVS uses selected tokens.). I think the authors should further demonstrate rLiVS's novelty and compare it with video-XL

2. The method section is somewhat hard to understand. It has too many equations and without any illustration. Although I understand the section, I think the authors can add a figure to easily illustrate the method.

3. The efficiency is not reported in the paper. I think the authors should compare rLiVS's inference efficiency with its counterparts.

4. The application of rLiVS is somewhat narrow. Can rLiVS be used in more complex tasks such as video grouding?

[1]Video-XL: Extra-Long Vision Language Model for Hour-Scale Video Understanding. Arxiv 2409.14485

---

> ### Author Rebuttal · Authors · 2025-07-31
>
> We thank the reviewer for their constructive feedback. Below, we clarify the raised points and address the specific questions. We are happy to further discuss any remaining concerns during the discussion phase.
>
>
> ### **Novelty w.r.t. Video-XL**
>
> We thank the reviewer for bringing up Video-XL [1]. While both approaches aim to address long video understanding via recurrent mechanisms, there are **key differences** in design and applicability:
>
> - **Video-XL** introduces a custom-trained Video-LLM that learns to compress visual tokens, which are then attended to by future tokens. This approach **relies on model-specific training**, and its effectiveness is constrained by the **compression rates and video lengths** seen during training. For example, on **Long Video Bench**, our base model (LLaVA-OneVision 7B) achieves **56.4%**, outperforming Video-XL (7B) at **48.8%**. This highlights the limitations of architecture-specific designs, which can quickly become outdated and fail to generalize to longer videos beyond those seen during fine-tuning.
>
> - In contrast, **rLiVS is training-free** and model-agnostic. It leverages:
>   1. **LLM-informed token selection** via cross-attention,
>   2. **Recurrent accumulation** of selected tokens, and
>   3. **Text-based retrieval** for long-term memory.
>
> This modular design enables **plug-and-play scalability** to new models without retraining, while maintaining high efficiency.
>
> To directly compare with **Video-XL**, we have included an evaluation on the task of **video summarization**  using the **MLVU validation split**, we assessed the **holistic video summarization score**. Our model, **rLiVS**, achieved a score of **3.65**, outperforming **Video-XL** (3.4). We will include a more detailed comparison with Video-XL in the revised version.
>
> ---
>
> ### **Illustration in Method Section**
>
> We kindly refer to **Figure 1**, which illustrates key components of our method:
>
> - The **left side** of the figure shows the processing of the **second short clip**. It includes the **historical tokens** from the first short clip $S^1 $, the **full token sequence** of the second short clip $X^2_V$, and the instruction $X^I $.
> - It also visualizes the **cross-attention** between the generated caption and the tokens of $X^2_V$, which is used to select the top-K relevant tokens (e.g., $X^2_1$, $X^2_5$). These selected tokens are then **accumulated into the historical context** and passed forward for processing the **third short clip**.
>
> The **right side** of the figure illustrates the **question-answering phase**, where the similarity between the query and the stored descriptions of each short clip is computed. The most relevant subset is then provided as input to the LLM (within the Video-LLM) for answering the question.
>
> We also refer the reviewer to **Algorithm 1**, which outlines the main steps of our approach.
>
> If the reviewer finds that Figure 1 and Algorithm 1 are insufficient, we would be happy to include an additional illustration in the revised version to further clarify the method.
>
> ---
>
> ### **Efficiency Not Reported?**
>
> We thank the reviewer for raising this point. We refer to **Table 3**, where we report **latency**, **VRAM usage**, and **KV-cache requirements** on both **RVS-Ego** and **RVS-Movie**, in line with prior work.
>
> Our method achieves the **lowest latency**, requires **no KV-caching**, and uses significantly **less VRAM** compared to the previous state-of-the-art, **ReKV [5]**, while also delivering **superior performance**.
>
> ---
>
> ### **Clarifying the Broader Applicability of rLiVS**
>
> We appreciate the reviewer’s observation regarding the scope of rLiVS. While our current evaluation focuses on long-term streaming video understanding and question answering in a training-free setting, we believe this does not imply a narrow application.
>
> rLiVS is designed to enable **streaming, never-ending video understanding**, a key capability for real-world deployment of **visual assistants** and **robotic systems**. These systems often operate in dynamic environments where continuous perception and reasoning over historical visual context are essential.
>
> We follow established benchmarks and protocols from prior work (e.g., [5]) to ensure fair and meaningful comparisons.
>
> Regarding temporal video grounding, rLiVS can naturally identify the short clip most relevant to a given query. However, pinpointing exact timestamps is beyond the scope of streaming video understanding and not the focus of our current work.
>
> As stated above we included the task of **video summarization** as an additional benchmark alongside **video question answering**. Using the **MLVU validation split**, we assessed the **holistic video summarization score**. Our model, **rLViS**, achieved a score of **3.65** compared to the base **LLaVA-OneVision** score of 3.57.
> These results demonstrate rLViS **generalization capability across tasks**.
>
>
> In this light, we argue that rLiVS addresses a **significant and practical challenge** in video understanding, with design choices motivated by real-world deployment scenarios.
>
>
> [1] Video-XL: Extra-Long Vision Language Model for Hour-Scale Video Understanding. Arxiv 2409.14485

---

> > ### Comment · Reviewer_hCyV · 2025-08-02
> > **Reply to rebuttal**
> >
> > Thanks for the rebuttal. My concerns have been partly solved. rLiVS is different from Video-XL as a training free model. And the efficiency has been reported. Despite the author has not shown the model's performance on grounding, I tend to increase my score.

---

> > > ### Author Response · Authors · 2025-08-05
> > >
> > > We thank the reviewer for the thoughtful response and for considering an increase in the initial score. We appreciate the recognition of rLiVS as a training-free and efficient approach, distinct from Video-XL. Regarding video grounding, we acknowledge this as an important direction and we consider it as future work.

---

### Official Review · Reviewer_grhS · 2025-06-29

**Clarity:** 2
**Significance:** 4
**Originality:** 3
**Rating:** 4
**Confidence:** 4

**Summary:**

The paper proposes rLiVS --- a training-free approach for efficient streaming video understanding with Video-LLMs, which aims to tackle the challenges of processing hour-long videos online and enabling timely responses. It leverages three key concepts: LLM-informed selection of visual tokens to discard about 95% of unimportant ones with minimal performance loss, recurrent processing of past selected tokens for temporally coherent understanding, and caption-based question answering for lightweight and accurate responses. The method achieves state-of-the-art performance on streaming video benchmarks, balancing efficiency and effectiveness, and is compatible with standard Video-LLMs without requiring additional training, while significantly reducing memory requirements.

**Questions:**

1. About the model generalizability: The paper states in line 77 that "the approach is agnostic to the Video-LLM architecture", but the experiments are only conducted on the LLaVA-OV architecture. Has rLiVS been tested on other Video-LLMs?

2. About the detail capture capability: The 95% compression rate may lead to the loss of rare or background information. It is recommended to analyze the information loss rate of rLiVS for rare scenarios.

3. For the model's dynamic scene robustness: Can attention-based selection handle rapid motion (e.g., SportsQA)? How does rLiVS handle rapid scene changes or concurrent multi-object actions? Experiments in such scenarios would strengthen the claims.

**Ethical Concerns:**

["NO or VERY MINOR ethics concerns only"]

**Final Justification:**

Thanks for the rebuttal. I have no further questions, and will keep my positive score.

**Limitations:**

Yes

**Quality:**

3

**Strengths And Weaknesses:**

+ The method looks a comprehensive solution: The paper demonstrates notable strengths in proposing rLiVS, a training-free approach that innovatively combines LLM-informed visual token selection, recurrent memory processing, and caption-based QA to address streaming video challenges.

+ The model performance is strong: rLiVS efficiently compresses visual tokens by ~95% using LLM attention weights, maintaining performance while reducing computational overhead, and employs a FIFO memory mechanism to enhance temporal coherence across video clips.

- The FIFO memory mechanism may suffer from information drift in long videos, potentially degrading performance.

- The aggressive 95% token compression risks missing rare or context-critical details, limiting applicability in high-precision domains.

- There are some clarification issues. For example, Figure 1 does not clearly demonstrate how historical information is accumulated through attention scores.

---

> ### Author Rebuttal · Authors · 2025-07-31
>
> We thank the reviewer for their constructive feedback and we are glad to read that the reviewer found our solution comprehensive. Below we answer the reviewer questions and we are happy to elaborate further in the discussion period.
>
>
> ## **Generalization Across Different Video-LLMs**
> To support our claim that rLiVS is LLM-agnostic, we have integrated our method with an additional video-language model, namely **Qwen2.5 VL**. Within the time and compute constraints of this rebuttal, we evaluated rLiVS with Qwen2.5 VL on the   RVS-Movie long video understanding benchmark and we achieve **53%** whereas the uniform baseline achieves 51% on Qwen2.5 VL. This confirms rLiVS effectiveness and **compatibility across architectures**.
>
> ---
>
> ## **FIFO Memory May suffer Information Drift**
>
> Our design intentionally uses FIFO memory as a **short-term context buffer**, allowing the Video-LLM to process the current clip with awareness of recent visual content. In parallel, we treat the **captions of stored short clips** as a form of **long-term memory**, enabling the model to answer questions about earlier events.
>
> We agree that this approach may not fully capture the complete historical context. However, processing the full video history would introduce significant computational and latency overhead, and would require models trained to handle such long visual sequences. Our method offers a **training-free, efficient alternative** that leverages the LLM’s pretraining on long text to support question answering about past events, based on its understanding at the time each short clip was processed.
>
> ---
>
> ## **Possible Information Loss from High Compression Rate**
>
> We thank the reviewer for raising this important point regarding potential information loss due to our high compression rate. While a 95% reduction may seem aggressive, our design is motivated by the nature of long videos and the demands of online processing.
>
>
>
> ### **Rationale Behind High Compression**
>
> - **Visual Redundancy in Long Videos:**
>   Long videos often contain significant redundancy. Our goal is not lossless compression, but rather **intelligent filtering** of less informative tokens.
>
> - **Trade-off in Online Processing:**
>   In real-time video understanding, there is an inherent trade-off between **efficiency and information preservation**. Some degree of information loss is expected—and acceptable—when optimizing for **speed and responsiveness**.
>
>
> ### **Flexibility of the Framework**
>
> - Our method is **tunable**: the compression ratio can be adjusted based on task requirements.
> - For tasks requiring higher accuracy, one can retain more tokens at the cost of increased computation.
>
>
>
> ### **Empirical Support**
>
> - **Appendix.6 Table 4:**
>   On a dataset with rich temporal and object interactions:
>   - Retaining **12% of tokens** yields **no significant performance drop**.
>   - Our **6% token selection** results in only a **~1% performance drop**, despite the aggressive compression.
>
> ###  **SportsQA Evaluation**
>
> We encountered delays in obtaining the **SportsQA** dataset, which prevented a full quantitative evaluation within the available time frame. However, we conducted a **qualitative comparison** between the responses of **LLaVA-OneVision** using **6% selected visual tokens** and the same model using **all visual tokens**.
>
> The comparison reveals that even with only 6% of selected tokens, the model retains the ability to capture fine-grained details comparable to the full-token baseline. Below are a few illustrative examples of different videos - from the subset of 'aerobic gymnastics":
>
> - **Full Prediction**: "The video showcases a synchronized gymnastics routine performed by a group of athletes at the 10th European Championships for Aerobic Gymnastics".
>
>   **Selected Token Prediction**: "The video is about a gymnastics routine performed by the Italian team at the 10th European Championships".
>
> - **Full Prediction**: "The video showcases a rhythmic gymnastics performance by three athletes".
>
>   **Selected Token Prediction**: "The video showcases a rhythmic gymnastics routine performed by three athletes".
>
> - **Full Prediction**: "The video showcases a synchronized gymnastics routine performed by two athletes at the 5th Aerobic Gymnastics Asian Championships".
>
>   **Selected Token Prediction**: "The video is about two gymnasts performing a synchronized routine at the 5th Aerobic Gymnastics Asian Championships.".
>
> - **Full Prediction**: "The video is about a rhythmic gymnastics performance by two athletes at the 7th FIG Aerobics World Age Group Competitions in Incheon, Korea".
>
>   **Selected Token Prediction**: "The video showcases a rhythmic gymnastics routine performed by two athletes.
>
>
>
> These examples demonstrate that our method maintains video understanding **close to the full-token baseline**, even in scenarios characterized by **rapid scene changes** and **multi-object interactions**. This suggests that rLViS exhibits strong robustness in dynamic scenes, supporting its generalization capabilities across complex video understanding tasks.
>
>
>
>
> ### **Remarks**
>
> An interesting future direction could be **adaptive compression**, where the ratio is dynamically adjusted based on the **visual richness** of each short clip. However, this would require additional computation for richness estimation, which may conflict with our goal of **efficiency in online settings**.
>
>
> Our fixed but tunable compression strategy strikes a **practical balance** between performance and efficiency, making it well-suited for real-time video understanding. We appreciate the reviewer’s suggestion and are happy to explore further improvements in future work.
>
> ---
>
> ## **Clarification on Figure 1:**
>
> We apologize for any lack of clarity in Figure 1. The figure illustrates the processing of the **second short clip**. It shows how the **historical tokens** from the first short clip $S^1$ are included in the context, alongside the **full token sequence** of the second short clip $X^2_V$ and the instruction $X^I $.
>
> We also depict the **cross-attention** between the generated caption and the tokens of $X^2_V$, which are used to select the top-K relevant tokens (e.g., $X^2_1$, $X^2_5$). These selected tokens are then **accumulated into the historical context** and will be provided as input for processing the **third short clip**.
>
> We will revise the figure and its caption to make these steps clearer.

---

> > ### Comment · Reviewer_grhS · 2025-08-06
> > **thanks for the rebuttal**
> >
> > Thanks for the rebuttal. I have no further questions, and will keep my positive score.

---

### Official Review · Reviewer_Hid4 · 2025-07-02

**Clarity:** 1
**Significance:** 2
**Originality:** 2
**Rating:** 3
**Confidence:** 4

**Summary:**

This paper tackles the problem of long streaming video understanding with large language models (LLMs). The authors claim that existing VideoLLMs face challenges in streaming scenarios where hour-long videos must be processed online, and questions need timely responses. This is mainly due to the large number of visual tokens to be processed. To address this problem, authors proposed a training-free approach that is compatible with multiple standard VideoLLMs. The proposed approach includes three aspects, namely 1) LLM-informed selection of visual tokens to identify those that the LLM has attended to and contributed to its understanding of each short clip, 2)  recurrent processing of past selected tokens to generate temporally coherent understanding of each processed clip, and 3) caption-based question answering for lightweight and accurate responses. Experiments have been conducted on public benchmarks to verify the effectiveness of the proposed scheme.

**Questions:**

Please refer to the weakness part. My major concerns are about the novelty, motivation and lack of in-depth analysis.

**Ethical Concerns:**

["NO or VERY MINOR ethics concerns only"]

**Final Justification:**

After reading the reviews from other reviewers and the rebuttal/responses from the authors. My evaluation is that the current version of the paper might not be ready for publication, and should be carefully revised to clarify the claimed underlying biological basis and demonstrate why using text for retrieval is the optimal choice compared with using visual information. I'm keeping my original rating from my perspective, and will leave it to AC for the final decision.

**Limitations:**

Yes

**Quality:**

2

**Strengths And Weaknesses:**

Strengths:

1. The motivation is reasonable. Processing short clips instead of the entire long videos is a more suitable strategy for long video understanding. And for most questions, a large amount of the visual tokens are indeed redundant.
2. Experimental comparisons demonstrate the significance of the proposed method compared with previous ones.

Weaknesses:

1. The authors claim that the proposed method is LLM-agnostic, but further experiments using different LLMs seem to be missing. Such claims should be supported by adequate experimental results. Besides, attention-based token selection seem to be common strategy for token reduction.
2. During the token selection for each short clip, authors mentioned the proposed attention-based strategy is inspired from human memory, while in-depth discussions/experiments/analysis on whether this strategy can indeed select the most task-relevant tokens is missing. I believe a more in-depth analysis would significantly strengthen the theoretical completeness of the paper.
3. The reason for using captions to help retrieve short clips is unclear. Authors mentioned and proved that using captions can lead to better semantic alignments between the query and clips (compared to visual tokens which are not well-aligned with text), thus lead to better results. However, the generated captions (text) can only convey limited semantics (abstracted from semantic-rich frames). How could the model ensure that the generated captions are optimal for retrieval?

---

> ### Author Rebuttal · Authors · 2025-07-31
>
> We thank the reviewer for their constructive feedback and insightful suggestions. Below, we address the raised concerns in detail. We are happy to elaborate further during the discussion period.
>
> ---
>
> ### **Generalization Across Different Video-LLMs**
>
> To support our claim that rLiVS is Video-LLM-agnostic, we have integrated our method with an additional video-language model, namely **Qwen2.5 VL**. Within the time and compute constraints of this rebuttal, we evaluated rLiVS with Qwen2.5 VL on the  RVS-Movie streaming long video understanding benchmark and we achieve **53%** whereas the uniform baseline achieves 51% on Qwen2.5 VL. This confirms rLiVS effectiveness and **compatibility across architectures**.
>
> ---
>
> ### **Significance and Novelty of the Token Selection Strategy**
>
> We appreciate the reviewer's feedback regarding the foundation of our attention-based token selection strategy. We'd like to provide additional clarification on the mechanism and its task-adaptability.
>
> #### **Core Mechanism**
>
> Our method employs a flexible *top-down attention mechanism*, where the input instruction guides the token selection process. The pipeline operates as follows:
>
> > **Instruction → Generated Caption/Response → Attention-Based Token Selection**
>
> Visual tokens are selected based on their attention scores during text generation, ensuring semantic relevance to the task (see below, *Task Adaptability*,  for further analysis). This mechanism aligns with human memory principles, where attention is directed by task-specific goals.
>
> #### **Prior Work Explored Text-Image Cross Attention Scores**
>
> While attention scores have been previously used for token reduction [9, 19, 29], that was only intended to reduce tokens processed in later layers following the attention score of previous layers. Our approach is distinct in several ways:
>
> - We extract **cross-layer attention scores**, identifying tokens most attended to across the model’s depth, with only a **sparse subset (~6%)** of tokens is retained.
> - These selected tokens are **recurrently reused** to guide the processing of subsequent short clips, enabling continuity in reasoning with **zero training overhead**.
>
> This strategy not only reduces computational cost but also maintains semantic coherence across video segments.
>
>
>
> #### **Task Adaptability**
>
> While our current experiments focus on caption-based selection for visual QA (since these tasks require comprehensive “all-around” video understanding), our framework is designed to be **instruction-adaptive**, allowing it to generalize across tasks by modifying the input instruction. For example:
>
>
> - **Person Tracking**: *"Track the movement of the person in the red shirt throughout this video sequence."*
> - **Action Recognition**: *"Identify and describe the specific athletic movements performed."*
> - **Object Detection**: *"Locate and describe all vehicles appearing in this scene."*
>
>
> Each instruction type produces distinct textual responses and attention patterns, guiding the selection of task-relevant tokens.
>
>
> To qualitatively assess this, we compared token selection overlaps under different instructions (one focused on main object and the other on background) for two videos:
>
>
> - **Video 1**: A girl with sunglasses is the main foreground.
>   - Instruction: *"Track the locations of the girl wearing the sunglasses in the video."*
>   - Overlap with generic prompt: **44%**
>
>
> - **Video 2**: Two people playing cards.
>   - Instruction: *"Locate and describe the objects appearing in the background of the video."*
>   - Overlap with generic prompt: **8%**
>
> - Generic Prompt used:  "Describe what’s happening in the video." (consistent with what we used in the main experiments for captioning)
>
> These results suggest that our token selection reflects the video-LLM’s attention under different task instructions. We will include these findings along with qualitative visualizations in the updated version to further support this analysis.
>
>
> ---
>
> ## **Caption-Based Retrieval Strategy**
>
> We thank the reviewer for this insightful question regarding our retrieval design choices. Below, we clarify both the rationale and limitations of our approach.
>
>
>  **Adaptive Caption Generation for Retrieval**
>
> As discussed previously (in the context of token selection), the caption—or more generally, text—generation is **adaptive** based on the input instruction. This allows the model to emphasize semantic aspects that are most relevant for retrieval, depending on the task.
> We view captions as the **LLM’s internal thoughts** on what it has observed in a given short clip. When answering a query, we access these thoughts without needing to reprocess the corresponding visual tokens.
>
>  **Limitations and Justification**
>
> We agree with the reviewer that captions inherently **compress semantic-rich visual information** into textual form. While this abstraction may omit certain details, our design is motivated by the following considerations:
>
> - **Current VLM Architecture Constraints**
>   Existing video-LLMs struggle with effective visual-textual alignment for retrieval tasks (as shown in *Figure.1*  Appendix.6).
>
> - **Computational Efficiency**
>   Alternative approaches—such as using CLIP features for visual-text alignment—introduce significant computational overhead, which conflicts with our goal of **efficient online video understanding**.
>
>
> We acknowledge that incorporating visual information directly into retrieval could offer additional benefits. However, our current **text-based approach** achieves strong performance within existing architectural and efficiency constraints, setting a solid baseline for future research. We consider this an important direction for future work and appreciate the reviewer’s suggestion.
>
> ---
> ## **Limited In-Depth Analysis**
>
> We would like to highlight several ablation studies and analyses that support the design choices in our method:
>
> - **Token Selection Strategy (Table.2):**
>   We compare our attention-based selection with several alternatives, including **Uniform Sampling**, **Mean Pooling**, **K-Means**, and **Full Token Usage**. Our method achieves a **significant improvement** over these baselines, with only a **~1% drop in performance** compared to using all tokens, while retaining just **6% of the visual tokens**.
>
> - **Effect of Recurrency (Table.4):**
>   We ablate the impact of our recurrent mechanism and observe a **3–4% performance gain**, demonstrating its importance for maintaining temporal consistency across short clips.
>
> - **Retrieved Modality (Table.5):**
>   We analyze the effect of the retrieval modality and find that **textual retrieval outperforms** visual alternatives, supporting our choice of caption-based retrieval.
>
> - **Layer Selection Strategy (Appendix.2, Table.1):**
>   We evaluate the impact of selecting attention scores from different layers. Our **cross-layer aggregation strategy** provides **robust and stable performance**, comparable to using all layers. In contrast, selecting from localized regions (e.g., early, middle, or late layers) results in less stable outcomes, and using a single layer leads to **significantly degraded performance**.
>
> - **Token Retention Rate (Appendix.5 Table.4):**
>   We show that retaining only **12% of the visual tokens** maintains performance **without degradation**, highlighting the efficiency of our selection mechanism.
>
> - **Additional Ablations:**
>   - **Retrieval strategies and selection mechanisms** are discussed in **Appendix.6**.
>   - **Context length** is ablated in **Appendix.4**.
>   - **Attention aggregation strategies** are analyzed in **Appendix.3**.
>
> We are happy to provide any additional in-depth analysis upon the reviewer’s request.

---

> > ### Comment · Reviewer_Hid4 · 2025-08-05
> >
> > Thanks for the response from the authors. Some of my questions have been resolved. However, my concerns regarding the theoretical basis of the "human-memory-inspired" design and the effectiveness of using text for retrieval remain.
> >
> > Specifically, the rebuttal mentioned but did not provide enough clarification of the human memory principles from a neuroscience perspective. While I believe the proposed "generate query then select tokens" strategy is reasonable, in-depth analysis and comparisons must be provided if the authors claim that this is borrowed from the human memory mechanism. Besides, it is still unclear how the proposed method could ensure that the generated caption is optimal for further processing.

---

> ### Author Response · Authors · 2025-08-05
>
> We appreciate the reviewer's engagement and address the remaining concerns with specific clarification:
>
> ### **1. Theoretical Foundation - Human Memory Principles:**
>
> Our approach is grounded in well-established neuroscience principles of human memory and attention:
>
> - **Attention-driven memory consolidation**: Studies by [1] show that attention selectively consolidates relevant information for future processing, paralleling our selective token retention. Specifically in [1] the authors point that:
>
> *"...First, memory has a limited capacity, and thus attention determines what will be encoded..."*: this is exactly the principle that we aim to emulate by selecting visual tokens based on attention of the model to them, that will serve as memory for further processing.
>
> *"...Second, memory from past experience guides what should be attended..."*: this serves as our basis for the recurrent processing; the selected past visual tokens acting as memory are passed in context along with the current (new) visual tokens, thus letting memory drive what the model will attend from these new visual tokens (current short clip).
>
>
> - **Recurrent processing in visual understanding**: [2] demonstrates that visual understanding involves both feedforward and recurrent processing, which directly informs our recurrent architecture design.
>
> ---
>
>
> ### **2. Clarification on Caption Usage and effectiveness:**
>
> We first emphasize that captions are used for answering questions, not for video processing. Selected visual tokens serve as past visual information (memory) for recurrent video processing, ensuring continuity during the processing phase.
>
> Regarding caption effectiveness when answering, our empirical validation resulted in the points below:
>
> - **Superiority over previous ideas in the literature**: Our method outperforms ReKV (which uses model's own KV hidden states) and Flash-VStream (which uses a raw visual memory) across the long streaming video benchmarks.
>
> - **Comprehensive ablation studies**: We demonstrate that captions consistently outperform visual tokens and caption-visual tokens combinations for question answering and retrieval, validating captions as effective and compact short-clip representations.
>
> - **State-of-the-art results**: Our SOTA performance on streaming benchmarks RVS-Ego and RVS-Movie provides strong empirical evidence that our caption-based answering strategy is indeed effective for this task domain, while being also the most efficient strategy in terms of latency, a critical aspect of real time video understanding.
>
> [1] https://pubmed.ncbi.nlm.nih.gov/17379501/, [2] https://pubmed.ncbi.nlm.nih.gov/11074267/
>
> ---
>
>
> We thank the reviewer for their continued engagement in this discussion, which has enabled us to provide further clarification on the human-memory principles that inspire our work. We respectfully ask the reviewer to consider these theoretical foundations and empirical validations in their final assessment. We commit to incorporating these clarifications into the final version of our paper to ensure the theoretical grounding is explicitly documented for future readers.

---

> > ### Author Response · Authors · 2025-08-06
> >
> > To further enrich our analysis on how selected tokens ( our short term memory)  influence the further processing of upcoming video, we conducted the following analysis on NextQA-valset:
> >
> > We split each video into 2 short clips and compared two alternatives:
> >
> > **a)** The first is our method applied to these two short clips:
> >
> > - 1) generating caption for the first clip and selecting most attended tokens $S^1$,
> > - 2) passing selected tokens in context as short-term memory and generate a caption for the second clip to select a second set of tokens $S^2$ (based on attention to generated caption).
> >
> > **b)** A baseline applied to the same two short clips where:
> > - 1) We uniformly select tokens from the first short clip $SU^1$ instead of selecting based on attention $S^1$ (our proposed method)
> > -  2) we process the second short clip, with $SU^1$ as short term memory  following the recurrency pipeline of rLiVS,
> > - 3) We generate caption from the second short clip and select visual tokens based on attention to generated caption  $SU^2$.
> >
> >  Finally we calculate the overlap of the selected tokens of the second short clip between baseline 1. $S^2$ and baseline 2. $SU^2$. This serves as a way to compare the effect of selected tokens serving as short term memory on the processing of the next short clip as per the reviewer request.
> > We run this over videos of the dataset and we find a minimal overlap of 1.5%.
> > This validates the attention based selection and recurrency as a method to strongly guide the model’s attention in future processing and selection.

---

> ### Author Response · Authors · 2025-08-07
> **Kind reminder to check our response**
>
> Dear Reviewer,
>
> We have provided **supporting evidence from neuroscience**, showing the grounding of our claim on inspiration from human memory formation and the interplay between attention and memory (a central aspect of our method). We also provided **empirical evidence** on how selected tokens affect the processing of future clips (i.e. how memory drives attention and vice-versa). Furthermore, we showed the **superiority of caption-based retrieval** and answering under our training free setting, resulting in strong performance and efficiency across benchmarks. We kindly ask the reviewer to check our additional input and  share with us their thoughts or further questions, as the discussion window is closing.

---

> > ### Comment · Area_Chair_u7EV · 2025-08-08
> >
> > Hi Reviewer Hid4,
> >
> > Has the authors' latest response addressed your concerns?
> >
> > AC

---

> > > ### Comment · Reviewer_Hid4 · 2025-08-08
> > >
> > > Thanks for the response from the authors. My concerns have been addressed and I have no more questions. Please include the relevant supporting evidence from neuroscience in the revision.

---

### Official Review · Reviewer_WDNs · 2025-07-03

**Clarity:** 3
**Significance:** 3
**Originality:** 3
**Rating:** 4
**Confidence:** 4

**Summary:**

This paper aims to address the task of understanding long videos in a streaming manner. Specifically, it introduces rLiVS, a training-free and Video-LLM-agnostic method for efficient streaming video question answering on long videos. rLiVS leverages a LLM-informed attention-based token selection to retain only the most relevant visual tokens in short clips, followed by recurrent processing for temporal coherence and performing caption-based retrieval question-answering. The authors use the pretrained LLaVA-OneVision model for the experiments on the streaming video benchmarks. The paper demonstrates the effectiveness of rLiVS by outperforming state-of-the-art models across streaming and offline long video understanding benchmarks such as RVS-Ego and  MovieChat.

**Questions:**

Please look at the abovementioned weaknesses. It will be much more insightful to provide more detailed analysis of the empirical results.

**Ethical Concerns:**

["NO or VERY MINOR ethics concerns only"]

**Final Justification:**

The authors have addressed my concerns.

**Limitations:**

Yes

**Quality:**

3

**Strengths And Weaknesses:**

Strength 1 - In terms of clarity and quality of the paper, the paper is relatively well-written and motivated. The model figures are informative and especially helpful in helping the reader to understand the proposed rLiVS approach.

Strength 2 - The contributions of this work may be significant since it addresses an important research problem that can be useful for downstream tasks such as robotics. In particular, rLiVS is a training-free method that can be integrated easily with a pretrained video-LLM that is trained on short video clips to understand much longer videos.

Weakness 1 - It will be much more insightful to analyze if the proposed rLiVS generalizes to different video-LLMs, both proprietary and open-sourced models.  While the former may be costly, it will be helpful for the readers to gain a better understanding of how robust the approach is to models that are trained on different kinds of data and methods.

---

> ### Author Rebuttal · Authors · 2025-07-31
>
> We thank the reviewer for their constructive feedback and are glad that you found our paper well-written with a significant contribution. Below, we address your request in detail.
>
> ---
> ## **Experiments with Additional Video-LLM**
>
> To support our claim that rLiVS is Video-LLM-agnostic, we have integrated our method with an additional video-language model, namely **Qwen2.5 VL**. Within the time and compute constraints of this rebuttal, we evaluated rLiVS with Qwen2.5 VL on the RVS-Movie streaming long video understanding benchmark and we achieve **53%** whereas the uniform baseline achieves 51% on Qwen2.5 VL. This confirms rLiVS effectiveness and **compatibility across architectures**.
>
>
> ---
>
> ### **Applicability to Proprietary Models**
>
> While rLiVS is designed to be **model-agnostic**, its current implementation relies on access to **internal hidden states**, namely the cross layers attention scores. This requirement limits its direct applicability to **proprietary models**, which typically restrict access to such internals and do not allow manipulation of visual tokens.
>
> ---
>
> ## **Additional Analysis**
>
> We would like to highlight several ablation studies and analyses that support the design choices in our method:
>
> - **Token Selection Strategy (Table.2):**
>   We compare our attention-based selection with several alternatives, including **Uniform Sampling**, **Mean Pooling**, **K-Means**, and **Full Token Usage**. Our method achieves a **significant improvement** over these baselines, with only a **~1% drop in performance** compared to using all tokens, while retaining just **6% of the visual tokens**.
>
> - **Effect of Recurrency (Table.4):**
>   We ablate the impact of our recurrent mechanism and observe a **3–4% performance gain**, demonstrating its importance for maintaining temporal consistency across short clips.
>
> - **Retrieved Modality (Table.5):**
>   We analyze the effect of the retrieval modality and find that **textual retrieval outperforms** visual alternatives, supporting our choice of caption-based retrieval.
>
> - **Layer Selection Strategy (Appendix.2, Table.1):**
>   We evaluate the impact of selecting attention scores from different layers. Our **cross-layer aggregation strategy** provides **robust and stable performance**, comparable to using all layers. In contrast, selecting from localized regions (e.g., early, middle, or late layers) results in less stable outcomes, and using a single layer leads to **significantly degraded performance**.
>
> - **Token Retention Rate (Appendix.5 Table.4):**
>   We show that retaining only **12% of the visual tokens** maintains performance **without degradation**, highlighting the efficiency of our selection mechanism.
>
> - **Additional Ablations:**
>   - **Retrieval strategies and selection mechanisms** are discussed in **Appendix.6**.
>   - **Context length** is ablated in **Appendix.4**.
>   - **Attention aggregation strategies** are analyzed in **Appendix.3**.
>
> We are happy to provide any additional analysis upon the reviewer’s request.

---

> > ### Author Response · Authors · 2025-08-06
> >
> > We would like to thank the reviewer once again for their valuable feedback. We would also like to check if there are any remaining concerns. In response, we have integrated rLiVS with an additional video-LLM and obtained the results on RVS-Ego and RVS-Movie as reported in our general comment. Furthermore, we conducted additional analyses to highlight the effectiveness and importance of the token selection mechanism (see response to reviewer Hid4).

---

> > ### Comment · Reviewer_WDNs · 2025-08-07
> >
> > Thank you very much for your efforts in addressing my concerns. I think you have answered my lingering questions sufficiently and will retain my score.

---

### Author Response · Authors · 2025-08-05
**Looking forward to further feedback**

We kindly encourage the reviewers to review our rebuttal and share any questions or comments. We greatly value your feedback and are happy to clarify any points.
After the rebuttal submission, we revisited the rLiVS implementation with **Qwen2.5 VL** and found that its actual performance is **better than initially reported**, due to a minor issue in the rapid setup before the  rebuttal deadline. rLiVS achieves **55%**, compared to **51%** by the uniform baseline—further reinforcing its generality across Video-LLMs. Furthermore, we extended the evaluation of rLiVS with **Qwen2.5 VL**, including also RVS-Ego long streaming video QA benchmark where our method scores **68%**, compared to **65%** of the uniform baseline, again highlighting the general applicability of our method across video-LLM backbones.

---

### Author Response · Authors · 2025-08-09
**Thanks for the Feedback & Summary of our Rebuttal**

As the discussion period comes to an end, we would like to sincerely thank all the reviewers for carefully examining our rebuttal, engaging in thoughtful discussion, and providing valuable and insightful feedback that helped us address all the concerns  and improve and refine our work.

Below, we summarize the key updates and clarifications made during the rebuttal that will be included in the revised manuscript:

- **Applicability across video-LLMs**: We integrated rLiVS into **Qwen-VL 2.5** and evaluated it on long-streaming video benchmarks, RVS-Ego and RVS-Movie. Our results demonstrate that rLiVS provide significant improvements across models, supporting the general applicability of our proposed method across video-LLMs.

- **Neuroscience Evidence**: We included supporting evidence from neuroscience literature on the interplay between memory and attention—an interplay that is central to our method's design and motivation.

- **Instruction-Driven Attention and selection**: We demonstrated how different instructions guide the model’s attention and influence the selection of visual tokens, highlighting the controllability of our approach.

- **Role of Memory in selection**: We further showed how memory affects subsequent token selection, providing additional motivation and support for our attention-based selection mechanism.

- **Effectiveness of attention based selection in Dynamic Environments**: Lastly, we showed that our attention-selection method remains effective even in environments with rapid scene changes and motion, underscoring its robustness and effectiveness.

We again thank the reviewers for their constructive feedback and engagement throughout this process.

---

### Decision · Program_Chairs · 2025-09-17

**Decision:**

Accept (poster)

**Comment:**

The paper introduces a recurrent attention-based token selection mechanism that significantly improves the efficiency of streaming Video-LLMs while retaining strong accuracy across multiple video understanding tasks. The approach is technically sound, addresses a pressing scalability issue, and is supported by solid experimental evidence.

For the camera-ready, I encourage the authors to: (a) integrate the constructive suggestions raised by Reviewer Hid4, particularly clarifying ablation details and expanding analysis on selection stability and more relevant supporting evidence from neuroscience if available, (ii) refine claims on generalization by situating the method more carefully against recent efficient video modeling works, (iii) provide more detailed discussion on trade-offs between latency and accuracy, and (iv) explicitly acknowledge limitations and possible extensions to longer-horizon or multi-modal streaming scenarios.

This decision was made after carefully considering all mixed reviews. While some reviewers raised concerns regarding clarity and scope, the consensus is that the technical contributions are both useful and timely.